# Unveiling the MUFA–Cancer Connection: Insights from Endogenous and Exogenous Perspectives

**DOI:** 10.3390/ijms24129921

**Published:** 2023-06-08

**Authors:** Zhiqiang Guo, Karl-Frédérik Bergeron, Marine Lingrand, Catherine Mounier

**Affiliations:** 1Biological Sciences Department, Université du Québec à Montréal (UQAM), Montréal, QC H3P 3P8, Canada; 2Department of Biochemistry, McGill University, Montréal, QC H3A 1A3, Canada

**Keywords:** monounsaturated fatty acids (MUFAs), stearoyl-CoA desaturases (SCD), oleic acid (OA), cancer

## Abstract

Monounsaturated fatty acids (MUFAs) have been the subject of extensive research in the field of cancer due to their potential role in its prevention and treatment. MUFAs can be consumed through the diet or endogenously biosynthesized. Stearoyl-CoA desaturases (SCDs) are key enzymes involved in the endogenous synthesis of MUFAs, and their expression and activity have been found to be increased in various types of cancer. In addition, diets rich in MUFAs have been associated with cancer risk in epidemiological studies for certain types of carcinomas. This review provides an overview of the state-of-the-art literature on the associations between MUFA metabolism and cancer development and progression from human, animal, and cellular studies. We discuss the impact of MUFAs on cancer development, including their effects on cancer cell growth, migration, survival, and cell signaling pathways, to provide new insights on the role of MUFAs in cancer biology.

## 1. Introduction

The rise in overweight and obesity over the last decades has become a public health concern worldwide [1]. There is consistent evidence that a higher amount of body fat is associated with increased risk for several cancer types, including stomach, pancreatic, liver, colorectal, and breast cancers [2,3]. Obesity and cancer are linked in a complex and multi-factorial manner. Obesity is associated with several metabolic and hormonal alterations that increase the risk of cancer in patients. One main driver for obesity is believed to be an overall rise in caloric intake, through increased consumption of carbohydrates and fat [4]. Furthermore, it is becoming increasingly evident that the nutritional state of a patient plays a role in their response to cancer therapy [5]. In obesity, excess body fat leads to alterations in lipid metabolism, including increased levels of circulating lipids, such as cholesterol and triglycerides. These lipid metabolism states can contribute to the development of cancer by promoting inflammation, oxidative stress, insulin resistance and hormone imbalance and chronically activating growth factor signaling, which can all increase the risk of cancer [2,4,6,7,8]. Cancer cells derive most of their energy from the breakdown of lipids originating from de novo lipogenesis or the diet [9]. There are changes in lipid metabolism that can support the growth and progression of cancer cells. For example, cancer cells have increased levels of some lipids, such as cholesterol, phospholipids, and fatty acids (FAs), which can contribute to rapid cancer cell growth and tumor formation [10,11]. Additionally, cancer cells display altered levels of FA metabolism, which can support the development of resistance to chemotherapy and radiation therapy. As such, the modulation of lipid uptake and metabolism are gaining much interest in the field, and new cancer treatment strategies are expected to emerge from these studies [5,11,12,13,14].

### 1.1. Lipid Uptake and Metabolism in Cancer

Lipids encompass a heterogeneous group of biomolecules that serve multiple essential functions in biological systems, including constituting the structural basis of biological membranes and serving as signaling molecules and energy sources [10,11,15]. In mammals, the main lipid class of molecules comprises FAs, acylglycerols, phospholipids, sterols, and sphingolipids [11,15,16]. Endogenous lipogenesis and exogenous (dietary) uptake are the main lipid supply sources for either normal or cancer cells (see Figure 1). Firstly, except for liver and adipose tissue, most tissues possess little capacity for de novo FA synthesis and depend on FA uptake for their needs [17]. Circulating lipids provided by the liver or adipose tissues can be taken in through the receptor-mediated endocytosis of low- or very-low-density lipoproteins (LDLs/VLDLs). Lipids are also imported via specific transmembrane transporters, such as the fatty acid-binding proteins (FABPs) and CD36 FA translocase, as well as members of the FA transport proteins (FATP1–6) and solute carrier family 27 (SLC27A1–6) [17,18]. In addition, during de novo lipogenesis, FAs are synthesized from cytoplasmic acetyl-CoA. Citrate, produced in the mitochondrial tricarboxylic acid cycle, is the main source of acetyl groups for FA biosynthesis. Acetyl-CoA is activated by acetyl-CoA carboxylases (ACC1/2) to form malonyl-CoA, which can be subsequently condensed via several steps catalyzed by the fatty acid synthase (FASN) to form the 16-carbon saturated FA palmitic acid. Palmitic acid can then be elongated by FA elongases (ELOVLs) and/or desaturated by stearoyl-CoA desaturases (SCDs) or fatty acid desaturases (FADSs) to form unsaturated FAs, such as the 16- and 18-carbon monounsaturated fatty acids (MUFAs) palmitoleic acid and oleic acid (OA) [11,14,15,16,19,20].

Altered metabolism is one of the most prominent hallmarks of cancer. The most understood metabolic change in cancer cells is the Warburg effect, which is the use of fermentation, even in the presence of oxygen, to generate ATP. It is characterized by an increase in glucose uptake and consumption, a decrease in oxidative phosphorylation and the production of lactate. As a corollary to this metabolic modification, cancer cells use carbon from glucose to build other biomolecules instead of completely oxidizing it to carbon dioxide [19,21]. In rapidly proliferating cancer cells, among other metabolic fuels, fatty acids are also an important source of energy. Rapid cancer cell growth and tumor formation demand increased lipid metabolism to meet their energy needs [10,22]. In non-cancer cells, a balance is maintained between lipogenesis and lipid degradation. Meanwhile, in cancer cells, lipids derived from de novo lipid synthesis are an important source of energy, and therefore the expression and activity of enzymes involved in lipid synthesis and transformation are increased, making them more independent from externally provided lipids [23,24,25]. Moreover, rapid cancer cell proliferation and tumor formation also demand increased lipid metabolism to meet cell membrane synthesis needs [10,22]. Several studies have demonstrated that an increase in ATP citrate lyase activity (catalyzing formation of cytosolic acetyl-CoA from mitochondria-derived citrate) and ACC1/2 activity is found in many cancers, such as breast, liver, ovarian and colorectal cancer [26,27,28,29,30,31,32]. Similarly, FASN also shows increased expression in cancers such as breast and prostate cancer and correlates with poor disease prognosis [27,28,29,30,31,32]. The increase in de novo FA synthesis in cancer cells alters cellular lipid composition and can be used for diagnostics [33]. The limiting step in the synthesis of de novo MUFAs, SCD activity, has also been found to be elevated in cancer cells [29,34]. Thus, the proportion of MUFAs could also be used as an important biomarker in cancer screening [28,35,36]. In parallel to the role of lipogenesis, cellular FA uptake was also implicated in the progression of some carcinomas [29,30]. The relative contribution of de novo synthesis and uptake depends on the availability of different lipid species within the extracellular milieu. While this can be influenced by the lipid composition of the diet, heterogeneity in the tumor microenvironment, due to ongoing vascularization, also has a major effect on local lipid availability [10,11]. In addition to FA synthesis and uptake, altered lipid metabolism in cancer cells also impacts energy production. For instance, overexpression of fatty acid oxidation (FAO) enzymes has been observed in various cancer types [37]. Inhibition of FAO has been shown to reduce tumor growth in multiple experimental tumor models [10,11]. Certain enzymes involved in β-oxidation, such as α-methylacyl-CoA racemase (AMACR) and carnitine palmitoyl transferase 1B (CPT1B), are specifically upregulated in colorectal, hepatic, and prostate cancers, whereas CPT1A is elevated in breast cancer [37,38,39,40]. Furthermore, altered FA metabolism is also involved in oncogenic signaling, cancer epigenetic alterations, supporting tumorigenesis and cancer progression and driving cancer stem-like cell phenotypes (see review [11,40]). Considering the extensive roles of FAs in cancer pathogenesis via the interplay between oncogenic signaling and lipid metabolism, regulating processes involved in cancer cell growth, survival, dissemination and metastases formation, there is potential for treatment strategies that leverage the selective metabolic vulnerabilities caused by these changes [9,10,11,40]. It is worth highlighting that the scope in this review is to examine the metabolism of MUFA specifically in relation to cancer as the broader topic of lipid metabolism, and its association with cancer has been extensively covered in the existing literature. By focusing on MUFA metabolism, our aim is to provide valuable insights into its potential applications in cancer therapies and shed light on future research directions in this specific area.

**Figure 1 ijms-24-09921-f001:**
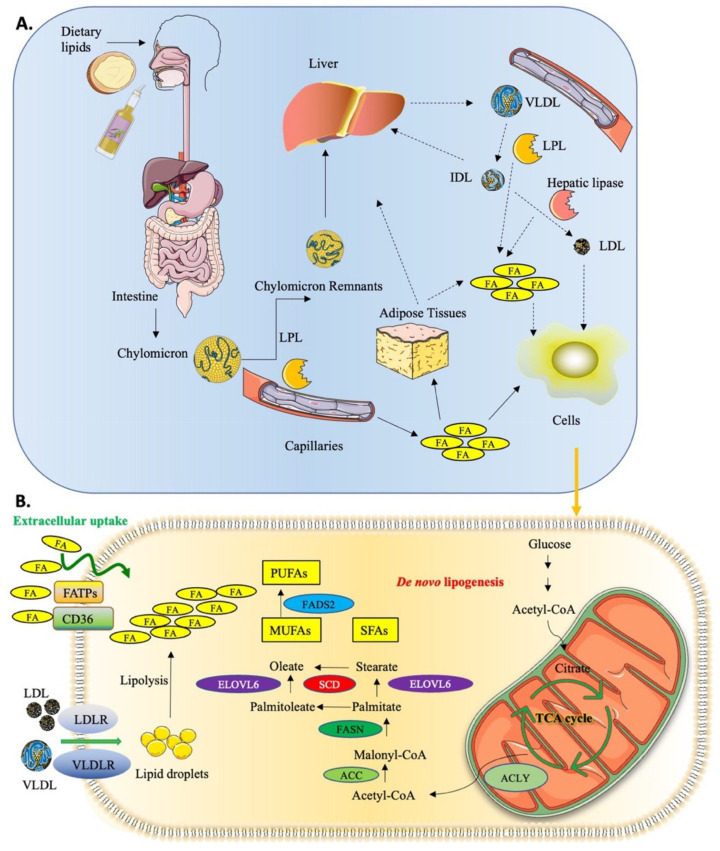
Dietary lipid uptake and lipid metabolism. (**A**) Dietary lipids are digested by the digestive system, absorbed in the intestine, and converted to triglycerides and cholesterol, which are then incorporated into chylomicrons. Lipoprotein lipase (LPL) hydrolyzes the triglycerides in chylomicrons to fatty acids (FA), allowing them to enter the lymphatic and circulatory system. The released FAs can be stored in adipose tissues or directly taken up by cells. The chylomicron remnants are cleared from circulation by the liver, which in turn releases very-low-density lipoproteins (VLDLs) into the circulation where they are hydrolyzed into intermediate-density lipoprotein (IDL) by LPL. IDL is then converted to low-density lipoprotein (LDL) by hepatic lipase or taken up by liver via the LDL receptor. Circulating lipoproteins and released FAs can be taken up by cells. Dotted lines indicate that endogenous lipid sources are involved. (**B**) At the cellular level, circulating FAs might enter the cell through simple diffusion and/or via some membrane transporters, such as fatty acid transport proteins (FATPs) and FA-binding protein CD36. Cells can internalize circulating lipoproteins via their cognate receptors (LDLR/VLDLR) and release FAs from them by intracellular lipolysis. FAs can also be biosynthesized intracellularly by de novo lipogenesis from acetyl-CoA produced by the mitochondrial tricarboxylic acid (TCA) cycle. ATP citrate lyase (ACLY) provides acetyl groups to the acetyl-CoA carboxylase (ACC) and the fatty acid synthase (FASN), allowing the synthesis of saturated fatty acid (SFA) palmitate. Palmitate can then be elongated by the fatty acid elongases (ELOVL) and/or desaturated by stearoyl-CoA desaturases (SCDs). Monounsaturated fatty acids (MUFAs) produced by SCD can be further desaturated by fatty acid desaturases (FADSs) to yield non-essential polyunsaturated fatty acids (PUFAs). The figure was created using Servier Medical Art image templates under a Creative Commons Attribution 3.0 Unported License.

### 1.2. MUFA Biosynthesis

The primary desaturases that are involved in the de novo synthesis of MUFAs in humans are SCDs, which are a family of enzymes localized in the membrane of the smooth endoplasmic reticulum (ER) [41,42]. They are essential enzymes for the survival of organisms, from bacteria to mammals [43,44,45,46]. There are five SCD isoforms (SCD1–5) in vertebrates [42]. In humans, only two variants exist: SCD1 and SCD5 [41]. The main isoform, SCD1, is expressed in most tissues [42] while SCD5 is mainly expressed in embryonic tissues but also in the brain and pancreas of adults [47]. The SCDs catalyze the formation of a double bond in the delta-9 position of saturated fatty acids (SFAs), creating a single unsaturation. The main products of SCD1, OA and palmitoleic acid, are formed from stearic acid and palmitic acid, respectively [47]. The regulation of human SCD expression and functional activity has been discussed in many comprehensive reviews (see [48,49]). Briefly, SCD1 expression is modulated by a variety of lipogenic transcriptional factors that bind to the gene promoters. Sterol regulatory element binding protein 1 (SREBP1) and carbohydrate response element binding protein (CREBP) act synergistically in the induction of *SCD1* expression (and other lipogenic genes) in response to insulin and glucose, respectively [50,51]. The regulation of expression is complexified by the binding of other transcription factors, such as PPARα, liver X receptor, CCAAT/enhancer binding protein α, nuclear transcription factor Y, neurofibromin 1 and specificity protein 1, all of which are activated by various growth factors, cytokines, hormones, and nutritional status [48,50]. Of note, phosphatidylinositol-3 kinase (PI3K)/Akt/mammalian target of rapamycin (mTOR) signaling was found to enhance SCD1 expression as part of the mechanism of lipogenesis activation in cancer cells [48,49,52,53]. Interestingly, *SCD1* is directly targeted by the tumor suppressor p53, suggesting that an increase in SCD1 expression and activity can be a key event in the development of cancer [48,49,54].

Despite the central role of canonical SCDs and their products OA and palmitoleate in MUFA studies, recent research has found other relatively rare MUFA isomers in certain tissues and cancer cells. These unusual MUFAs are catalyzed by other FADSs [55,56,57]. For example, a recent study identified elevated sapienate, desaturated from palmitate by FADS2, in some cancer cell lines [57]. Sapienate supported cancer cell membrane biosynthesis and proliferation in an SCD-independent way, which increased cancer plasticity [57]. An in-depth lipidomic study of prostate cancer cells revealed a diversity of unusual MUFAs, such as n-5, n-13, n-8, n-10, and n-12 FAs, which are related to FADS1/2 activity [55]. In addition, a lipidomic study in breast cancer cells reported that the inhibition of SCD1 led to an increase in n-10 MUFA isomers that depended on FADS2. Interestingly, high FAO activities were found in some specific subtypes of human breast cancer cell lines, which are correlated with cancer metastasis and invasiveness [56]. These results indicate that a diversity of alternative MUFA metabolic pathways is involved in lipid metabolism to promote cancer plasticity. The main biosynthetic pathways of MUFA are summarized in Figure 2.

## 2. Exogenous MUFA and Cancer

An understanding of the impact of MUFA on cancer progression must consider the role of circulating MUFA, most prominently derived from the diet. They are commonly found in foods, such as olive oil, avocados, nuts, and seeds, and have been the subject of extensive research due to their potential health benefits [11]. The common MUFAs and their outcomes on cancer have been summarized in Table 1.

### 2.1. Dietary MUFA and Cancer Risk—Evidence from Human Studies

OA is the most abundant MUFA in the human diet, accounting for around 20% of FAs in most fat sources. It is highly enriched in olive oil, where it reaches almost 80% of FAs [14,78]. High consumption of olive oil is a main feature of the Mediterranean diet, which is renowned for its health benefits and protective effect on cardiovascular diseases, diabetes, obesity, and cancer [78,79,80,81,82,83]. A recent comprehensive meta-analysis of 117 studies including 3,202,496 participants evaluated the association between the Mediterranean diet and cancer risk [81]. The highest adherence score to a Mediterranean diet was inversely associated with mortality in cases of breast, colorectal, head and neck, respiratory, gastric, bladder and liver cancers. However, the risk of blood, esophageal, pancreatic, and prostate cancers was not modified [80,81]. Furthermore, individual component analyses showed that the protective effects against cancer risk were mostly attributed to lower alcohol consumption, whole grain intake and fruit and vegetable intake. No clear association was identified for olive oil through this approach [80,81]. These results hint at a complex, multi-target impact of the Mediterranean diet, of which olive oil is only one component. To better understand the role of dietary MUFAs in cancer risk, we reviewed the recent epidemiological and clinical studies focusing on MUFA-enriched diets, mostly based on olive oil.

Several studies have shown that MUFA intake was associated with a decreased risk of cancer. A meta-analysis of 38 studies found that olive oil consumption was positively associated with lower odds of developing breast cancer and digestive cancers (colorectal, oral cavity, pharynx, esophagus, and pancreatic cancers) [83]. A randomized clinical trial performed in Spain found a significant inverse association between the consumption of a Mediterranean diet supplemented with extra-virgin olive oil and breast cancer incidence. A high consumption of extra-virgin olive oil (≥15% of total energy intake) is instrumental for obtaining this significant protection [84]. Similarly, another research study also reported an inverse association between OA intake and breast cancer [85]. In the European Prospective Investigation into Cancer and Nutrition cohort, dietary total MUFA was inversely associated with colon cancer but not rectal cancer [86]. A cohort study using 7-day food diaries in England assessed dietary OA in the etiology of pancreatic cancer. They found large, dose-dependent, inverse associations between OA intake and pancreatic cancer risk [87]. Another case–control study of 462 cases of pancreatic cancer and 4721 controls from eight Canadian provinces reported that dietary MUFAs were associated with a 28% reduced risk of pancreatic cancer [88]. Furthermore, the results from the French prospective cohort NutriNet-Santé showed that MUFA intake was associated with a decreased risk of digestive cancers (esophagus, liver, stomach, pancreas, and colorectal cancers) [89]. A New Zealand study showed that increasing the intake of MUFA-enriched vegetable oil was associated with a progressive reduction in prostate cancer risk [90]. In addition to olive oil, another study found that avocado intake, as a source of dietary MUFA, was associated with a reduced risk of prostate cancer [91]. Erucic acid (C22:1) is rich in the Chinese diet. This might contribute to the lower incidence of brain cancer in Chinese children as high levels of erucic acid have been found in the breast milk of Chinese women [73].

On the other hand, some studies have demonstrated either no correlation or an elevated risk of cancer development with MUFA consumption. A meta-analysis of 10 studies, including 8 case–control studies and 2 prospective studies, examined the association between olive oil intake and breast cancer risk. Although it suggested a potential inverse association between MUFA consumption and breast cancer, it was not statistically significant [92]. Similarly, other meta-analysis studies also reported that there was no significant positive or negative trend in breast cancer risk with dietary MUFA intake [85,93,94,95]. A Korean colorectal adenoma study found that there was no significant association with MUFA intake in adults [96]. Observational studies including 13 case–control studies and 7 prospective studies showed no significant difference between high versus low MUFA intake and pancreatic cancer risk [97]. No correlation between the intake of MUFA and pancreatic cancer was observed in a large cohort of US women during the subsequent 18 years of follow-up in the Nurses’ Health Study [98]. A case–control study from southeast China based on a questionnaire also failed to show a statistically significant association between MUFA intake (including C14:1, C18:1, C20:1, and C22:1) and the incidence of oral cancer [99]. Interestingly, in a hospital-based large-scale case–control study, the replacement of MUFAs with carbohydrates, SFAs and PUFAs for every 50 kcal of energy was associated with increased odds of breast cancer [100]. Dietary MUFA was also reported to be associated with an increased risk of pancreatic cancer in a case–control study from the San Francisco Bay and a large prospective cohort from the NIH-AARP Diet and Health Study [101,102]. A population-based cohort study performed on Chinese men showed that the dietary intake of MUFA was associated with an increased risk of liver cancer [103]. Lastly, a recent follow-up study from the Nurses’ Health Study and the Health Professionals showed a positive association between MUFA intake and colorectal cancer risk [104].

Overall, the evidence from human studies regarding the association between dietary MUFA and cancer risk is ambiguous, particularly in pancreatic and colorectal cancers where reported results are contradictory. One possible reason is that dietary MUFAs come from different sources, such as plant and animal sources, with the presence of additional other dietary components that might potentially obscure the associations between MUFAs and health outcomes [105]. In two large prospective cohorts of US men and women, total MUFAs and MUFA intake from plants were inversely associated with total mortality after adjusting for potential confounders, whereas MUFA intake from animal sources was associated with higher mortality [106]. A recent study also showed that the intake of MUFA tended to be positively associated with the risk of colorectal cancer while this positive association was mainly driven by dietary MUFAs coming from animal sources [104]. The specific FA composition may also influence the effects. For example, an increased risk of breast cancer was associated with increasing levels of the trans-MUFAs palmitoleic acid and elaidic acid while cis-MUFAs were unrelated to breast cancer risk [107]. A diversity of minor compounds is contained in dietary sources of MUFA. In olive oil, although OA is the primary component, there are other FAs and many minor compounds in the unsaponifiable fraction. Of note, some of them have been defined as “bioactive compounds” and have been shown to exert chemopreventative effects on cancer, such as hydroxytyrosol [108,109], oleuropein [109,110], oleanolic acid [111], oleocanthal [112] and pinoresinol [113]. Another study showed that OA and the representative minor components of olive oil have opposite effects. The treatment of colorectal cancer cell Caco-2 with OA (1–100 μM) induced DNA synthesis and cell growth, while minor compounds (hydroxytyrosol, oleuropein, pinoresinol, squalene and maslinic acid; 0.1–10 μM) reverted these effects. These results suggest that different sources of dietary MUFA, containing various minor compounds, can have different effects on cancer [114]. Lastly, the dosage of MUFA might also play an important role. For example, low OA concentrations increased Ca^2+^ entry (related to cell proliferation) while higher OA concentrations inhibited it in HT29 human colorectal adenocarcinoma cells [115]. In a commentary paper, the authors expressed their concerns that high concentration and long-time treatment with OA could lead to apoptosis [116]. The inconclusive results of the human studies described above highlight the need for more research in this area, using rigorous study designs and methodologies, to fully understand the potential relationship between dietary MUFA intake and cancer risk.

### 2.2. Exogenous MUFA and Cancer Risk—Evidence from Animal Models

As the evidence from human diet studies is inconclusive, animal studies could offer a more controlled environment in which to investigate the potential impact of MUFA consumption on cancer development. Access to animal tissues also allows for the investigation of the biological mechanisms involved. As such, experimental studies assessing the effects of dietary MUFA on cancer progression have been conducted in several animal models (see Table 2). These studies typically rely on feeding tumor-bearing animals a diet enriched with MUFA from sources such as olive oil. Occasionally, MUFA is directly injected into the animal, bypassing the digestive system.

MUFA-enriched diets have shown anti-cancer effects in animal models of colorectal and lung cancers. In a dextran sodium sulfate (DSS)-induced colon cancer mouse model, mice were put on 10% sunflower oil (SFO) or 10% extra virgin olive oil (EVOO) diets. EVOO-fed mice showed less incidence and multiplicity of tumors than SFO-fed mice. β-catenin immunoreactivity, proinflammatory cytokine production (TNF alpha, IL-6, INF gamma), cyclooxygenase-2 (COX-2) and inducible nitric oxidase synthase expression were significantly lower in the colon tissue of the animal group fed with EVOO than the SFO group, which is indicative of lowered inflammation and colorectal carcinogenesis progression [117]. In a dimethylhydrazine (DMH)-induced rat model of colon cancer, olive oil treatment lowered tumor incidence, multiplicity, and size, compared with treatment with DMH alone. Olive oil also reduced the expression of inflammatory and angiogenic markers (NF-κB, VEGF and MMP-9) and elevated the expression of pro-apoptotic markers (caspase-3 and -9) in DMH-treated rats [118]. In an azoxymethane/DSS-induced model of intestinal cancer on mice where the *Scd1* gene is specifically knocked out of intestinal tissue, an OA-rich diet reduced intestinal inflammation and significantly decreased the number and size of tumors [119]. In a murine lung adenocarcinoma LAC-1 transplantation mouse model, a diet enriched with olein (a palm oil fraction rich in OA) significantly delayed adenocarcinoma progression, increasing tumor latency and mice survival [120]. The effect of high-OA (C18:1) peanut oil and high-linoleic-acid (LA; C18:2) SFO was investigated in a mouse model of lung tumorigenesis (induced by a nicotine-derived NNK (N-nitrosamine: 4-(methylnitrosamino)-1-(3-pyridyl)-1-butanone). After 20 weeks of feeding, all mice fed the LA-enriched diet showed lung tumors (6.3 tumors on average per mouse). In comparison, the mice fed the OA-enriched diet presented a 25% lower incidence of lung tumors and a 31% reduction in the number of tumors per mouse, suggesting that OA specifically suppresses lung tumorigenesis in this model [121].

However, other studies have suggested that a high-MUFA diet may increase the risk of certain types of cancer. For instance, in a mouse model of pancreatic cancer, mice xenografted with HPAF cells were fed for 14 weeks with four different high-fat diets (15% fat, 4 kcal/g): SFA, MUFA, n-3 PUFA and n-6 PUFA. Except for the n-3 PUFA diet, which decreased tumor viability, mice fed with the other diets, including the MUFA diet (15% olive oil), showed an increase in tumor weight compared with an isocaloric control diet (5% fat, 4 kcal/g) [122]. In another study, nude mice implanted with cervical cancer cell HeLa were fed with a high-olive-oil diet (45% kcal fat). Compared to the control diet (10% kcal fat), the olive oil group showed a significant increase in tumor weight, by more than 6-fold. Xenograft tumor tissues from the olive oil group exhibited poor differentiation and higher heterogeneity. Immunohistochemistry analyses of these sections further uncovered a significant increase in cell proliferation (PCNA-positive cells) following olive oil treatment [123]. Another study from this group also showed that the high-olive-oil diet aggravated cervical cancer metastasis. They injected HeLa cells into the tail vein to cause metastasis in the liver. Mice in the olive oil group displayed a higher metastasis incidence and a significant increase in the size of the metastatic nodules, suggesting an association between dietary OA and cancer progression [124]. Lastly, a breast cancer study, using a female MMTV-neu(ndl)-YD5 transgenic mouse model (overexpression of *Erbb2*/*Neu*/*Her2*), compared the tumor effects of different dietary-FA-enriched diets: 10% safflower oil (n-6 PUFA), 3% menhaden oil + 7% safflower oil (marine-derived n-3 PUFA), 3% flaxseed + 7% safflower oil (plant-based n-3 PUFA), 10% olive oil (MUFA) or 10% lard (SFA). Marine n-3 PUFA best mitigated tumor outcomes, and MUFA, SFA and plant n-3 PUFA showed similar intermediary outcomes, while n-6 PUFA-fed mice had the poorest outcomes. Examination of tumor tissue revealed that the phospholipid fractions (phosphatidylcholine and phosphatidylethanolamine composition) were enriched in the FA families included in each experimental diet, suggesting that dietary FAs may exert their biological effects through cell-membrane-mediated mechanisms [125].

Intriguingly, there are a few studies testing the administration of refined OA to mouse models of cancer. In a tongue squamous cell carcinoma (TSCC) xenograft mouse model (CAL27 cells), intraperitoneal injection of OA had a marked inhibitory action on tumor growth. Immunohistochemical analyses of xenograft tumors showed that OA strongly inhibited p-Akt, p-mTOR and p-S6K expression and induced caspase-3 cleavage, indicating that OA could have valuable anticancer effects on TSCC via autophagy and apoptosis [126]. In a recent study, OA was incorporated into nanoparticles and either intravenously injected or administered by gavage to a breast cancer xenograft mouse model (4T1 cells). OA nanoparticles accumulated in tumors and triggered significant inhibition of tumor growth [127]. In a colorectal cancer xenograft mouse model (HCT116 cells), intragastric OA injection had no effect on tumor volume. However, tumor size was increased upon insulin injection, and this effect was potentiated by OA [128]. In another colorectal cancer xenograft mouse model (HC29 cells), nude mice were treated with elaidic acid and OA by gavage. The elaidic acid-treated group showed both increased subcutaneous tumor growth and metastases while the OA-treated group only showed increased peritoneal metastasis [129]. The enhanced metastasis results induced by elaidic acid could be attributed to the increased HT29 cell growth and stemness through the activation of EGFR in lipid rafts [129]. Furthermore, in a lung colonization model where head and neck squamous carcinoma cell TU183 were injected into the tail vein of mice, the preliminary injection of OA (mimicking high circulating free FAs) significantly increased the metastatic seeding of the lungs [130].

The results from animal studies presented here suggest a complex relationship between MUFA intake and cancer risk that might depend on a range of factors such as the type of tumor, the specific type and dose of MUFA consumed, as well as other aspects of the diet. The mixed results from the few studies using purified OA have so far failed to clarify this complexity.

**Table 2 ijms-24-09921-t002:** MUFA and mouse models of cancer.

Cancer Type	MUFA Source	Mouse Model	Main Outcome	Reference
Breast cancer	10% olive oil diet	MMTV-neu(ndl)-YD5 transgenic mouse model	Mitigated tumor outcome (though not as efficiently as a 3% menhaden oil + 7% safflower oil mix)	[125]
Breast cancer	OA nanoparticles, intravenous injection and gavage	Xenograft of 4T1 cells	Inhibited tumor growth	[127]
Cervical cancer	High-olive-oil diet (45 kcal % fat)	Xenograft of HeLa cells	Increased tumor growth	[123]
(Liver metastasis)		(Subcutaneous and tail vein injection)	Increased tumor metastasis	[124]
CRC	OA, injected intragastrically at a dose of 2.0 g/kg/day	Xenograft of HCT116 cells	No difference compared with controls	[128]
CRC	10% extra virgin olive oil diet	Chemically induced (DSS)	Reduced incidence and multiplicity of tumors	[117]
CRC	Olive oil 1 g/kg through oral gavage	Chemically induced (DMH)	Inhibited tumor growth	[118]
CRC	Fatty acid-rich diet, 75% OA	Chemically induced (AOM/DSS)	Reduced body weight loss and number and size of tumors	[119]
CRC	Oral intake of OA and elaidic acid	Xenograft of HT29 cells (subcutaneous, spleen, tail vein, and peritoneum)	Increased tumor growth and metastasis	[129]
HNSCC/Lung metastasis	OA, tail vein injection	Xenograft of TU183 cells (tail vein injection)	Induced metastasis	[130]
Lung cancer	6% OA-enriched diet	LAC1 tumor transplantation	Inhibited tumor growth but no impact on metastasis	[120]
Lung cancer	AIN-76A diet containing 10% OA	Chemically induced (NNK)	Reduced incidence and level of tumors	[121]
Pancreatic cancer	15% olive oil diet	Xenograft of HPAF cells	Increased tumor weight	[122]
TSCC	OA, injected intraperitoneally at 2/4 mg/kg	Xenograft of CAL27 cells	Reduced tumor volume and weight	[126]

CRC, colorectal cancer; HNSCC, head and neck squamous cell carcinoma; TSCC, tongue squamous cell carcinoma; DSS, dextran sulfate sodium; AOM, azoxymethane; LAC-1, lung adenocarcinoma 1; NNK, 4-(methylnitrosamino)-1-(3-pyridyl)-1-butanone.

### 2.3. Exogenous MUFA and Cancer Cell Behavior—Evidence from Cellular Models

In vitro cell culture models afford the possibility of direct exposure to known concentrations of specific MUFA. As such, this approach has yielded strong data characterizing the influence of these specific FA on cancer cell behavior. Furthermore, various signaling pathways have been identified that underlie the mechanisms of these effects. The well-delineated pathways triggered by OA are summarized in Figure 3.

#### 2.3.1. Effect on Cell Proliferation

Several studies have shown that OA significantly promotes the proliferation of breast cancer cells via signaling pathways dependent on the activation of G protein-coupled receptors (GPR) 40 and 120 [131,132,133]. In the triple-negative breast cancer cell line MDA-MD-231, OA binds to GPR40, which is coupled to G_i_/G_o_ and G_q_, and results in the activation of the PLC/PKC/Ca^2+^, PI3K/Akt and MEK1/2/Src pathways, promoting cell growth and proliferation [132]. These pathways are also implicated in promoting proliferation in prostate cancer cell lines PC3 and DU-145 [134]. GPR40 and GPR120 are expressed in poorly invasive MCF-7 breast cancer cells. The cell proliferation of these cells was also induced following stimulation with OA. This effect is dependent on Src kinase activation and EGFR transactivation, ERK1/2 phosphorylation on Thr-202 and Tyr-204 and the DNA binding of AP-1 [135]. In 786-O renal cell carcinoma cells, OA activates integrin-linked kinase (ILK) via GPR40, resulting in the activation of Akt and COX-2 and subsequently promoting cell proliferation [136]. Moreover, OA treatment stimulated cell proliferation in a dose- and time-dependent manner in the cervical cancer cell line HeLa. OA treatment increased the percentage of cells in the S phase, decreased cells in the G_2_ phase and increased the number of colonies in the colony formation assay. This proliferation effect is associated with CD36 upregulation, the best-characterized FA transporter. Inhibiting CD36 prevented the effect of OA on cell proliferation while overexpressing it mimicked the effects of OA [124]. Furthermore, OA activates Src kinase and the downstream ERK1/2-dependent signaling pathway in a CD36-dependent manner [124]. These results suggest that OA can promote cancer cell growth by inducing the expression of CD36, resulting in the activation of the Src/ERK signaling pathway [124].

#### 2.3.2. Effect on Cell Survival

In HepG2 hepatocellular carcinoma cells, OA was found to facilitate survival through the FABP5–hypoxia-inducible factor-1 alpha (HIF-1α) axis, which plays a pivotal role in response to hypoxic stress. Under hypoxic conditions and following OA exposure, HIF-1α was activated, and FABP5 was upregulated. OA treatment improved cell survival according to a colony-formation assay with an increased colony number and size. This phenomenon was suppressed when FABP5 or HIF-1α were silenced, indicating that the FABP5/HIF-1α axis is involved in OA-driven hepatocellular carcinoma cell growth [137]. In addition, a role for OA in prolonging breast cancer cell survival has also been described. OA can protect human breast cancer cells (MDA-MB-231 and MCF10A) against palmitate-induced apoptosis in part by increasing the esterification of this free FA (FFA) into triacylglycerol (TG), and OA can protect cells against apoptosis induced by serum withdrawal by the upregulation of the TG–FFA cycle [138]. Another study showed that OA treatment following serum deprivation specifically promotes cancer cell survival, growth, and migration in highly aggressive carcinoma cell lines, including gastric carcinoma cell HGC-27 and breast carcinoma cell MDA-MB-231, via AMPK activation [139]. In addition, in a co-culture system of adipocyte-breast cancer cells, OA secreted from adipocytes inhibited lipid peroxidation and the ferroptosis of triple-negative breast cancer cells [140]. In a recent study, OA treatment promoted H460 lung cancer cell survival under glucose-deficient conditions by activating lipid metabolism and inhibiting autophagy [141]. In esophageal squamous carcinoma cells, the high expression of the transcription factor BACH1 induced ferroptosis by inhibiting MUFA synthesis. OA significantly attenuated the ferroptosis phenotypes and reversed the cell death of BACH1-overexpressing cells. OA was found to be incorporated in the cell membrane and to protect the tumor cell from ferroptosis [142]. OA also plays a role in chemoresistance. OA treatment of PC3 and DU-145 prostate cancer cells interfered with the decline of cell viability induced by docetaxel, the first-line chemotherapeutic agent for the treatment of androgen-independent prostate cancer. This effect was mediated by the GPR40 receptor, suggesting that OA and GPR40 might represent a new prognostic factor and a molecular target for the treatment of advanced prostate cancer [134]. Another study found that elaidic acid, a trans form of OA, significantly enhanced the survival of CT26 and HT29 colorectal cancer cell lines. Elaidic acid enhanced cell proliferation and bestowed drug resistance to 5-fluorouracil, demonstrating tumorigenic potential [68]. Nervonic acid (C24:1), a long-chain MUFA produced by OA elongation, was also reported to protect PC-12 pheochromocytoma cells from oxidative stress [77].

#### 2.3.3. Effect on Cell Migration and Invasion

GPR40/120, EGFR and Akt-dependent pathways have been heavily involved in OA-induced migration in many cancer cell lines. In PC3 and DU-145 prostate cancer cells, OA was found to increase cell proliferation and migration via GPR40 and PI3K/Akt signaling [134]. In MCF-7 and MDA-MB-231 breast cancer cells, OA induced cell migration and invasion. Cell migration was dependent on GPR40/120, EGFR, PI3K and Akt activity, whereas invasion was mediated though PI3K and Akt. Furthermore, OA promoted the relocalization of paxillin to focal contacts in a PI3K- and EGFR-dependent manner [133]. Another study performed in MDA-MB-231 breast cancer cells found that OA induced MMP-9 secretion through a PKC, Src, and EGFR-dependent pathway, whereas it induced invasion via an EGFR, G_i_/G_o_ proteins, MMPs, PKC, and Src. In contrast, OA did not induce an increase in MMP-9 secretion in MCF10A and MCF12A mammary non-tumorigenic epithelial cells. This suggests that OA has an important role in the invasion process and metastasis in breast cancer [143]. In addition, an arachidonic acid (AA)-dependent pathway was implicated in OA-triggered breast cancer cell migration [144,145]. In MDA-MB-231 breast cancer cells, OA mediates the production of AA from membrane phospholipids through the activation of GPR40/120. AA metabolites then mediate focal adhesion kinase (FAK) phosphorylation and cell migration [145]. Free AA is metabolized by COX-2 and LOXs to produce eicosanoids. Eicosanoids bind and activate GPRs, which mediate EGFR transactivation and the activation of MMPs and Src. OA can then promote cell migration through the signal transducers and activators of transcription 5 (Stat5), of which the activation requires Src, MMPs, COX-2, and LOXs [144]. Furthermore, our recent study highlighted a phospholipase D2 (PLD2)/mTOR-dependent signaling pathway in OA-induced breast cancer cell migration. In wound healing assays, OA treatment increased the wound recovery of MDA-MB-231, T47D, and MCF-7 breast cancer lines. Analysis of migratory dynamics revealed that OA increased the speed and directionality of the migration of MDA-MB-231 cells. Further Transwell migration and invasion analysis showed that these changes were associated with the activation of PLD2 and mTOR [146].

In both colorectal cancer and head and neck squamous cell carcinomas (HNSCCs), OA was found to enhance cancer metastasis via angiopoietin-like 4 (ANGPTL4) pathways [130,147]. In HNSCC cell lines TU183 and HMEC-1, OA induced ANGPTL4 protein expression and secretion in a PPAR-dependent manner. The expression of ANGPTL4 induced epithelial–mesenchymal transition (EMT) markers vimentin, MMP-9, and fibronectin and its downstream effectors Rac1/Cdc42, which significantly promoted cell migration and invasion [130]. In SW480 colorectal cancer cells, OA promoted cell migration and invasion by the induction of NADPH oxidase 4 (NOX4), accompanied with increased levels of ROS and MMP-1/9. NOX4 induction and activation were ANGPLT4-dependent [147]. In addition, in 786-O renal carcinoma cells, OA was demonstrated to increase cell invasion in a dose-dependent manner, which was dependent on the ILK/COX-2/MMP-9 pathway [136]. A study using a two-dimensional co-culture system to simulate the crosstalk between adipocytes and gastric cancer cells showed that after co-culture with isolated omental adipocytes, gastric cancer cells exhibited significantly enhanced invasiveness. A lipidomic analysis showed that gastric cancer cells accumulated higher levels of OA during the co-culture. Further analysis in chick chorioallantoic membrane assays showed that OA treatment significantly promoted the invasiveness of gastric cancer cells and induced the expression of MMP-2 in gastric cancer cells by activating the PI3K/Akt signaling pathway in a PTEN-independent manner [148].

#### 2.3.4. Effect on Cancer Suppression

Though the studies presented above support the view that OA promotes cancer progression, several studies demonstrated a more complicated association between OA and cancer. In low metastatic carcinoma cells, such as gastric carcinoma cell SGC7901 and breast carcinoma cell MCF-7, OA inhibited cancer cell growth and survival [139]. Moreover, it was reported that a relatively high concentration (1 mM) of OA could inhibit DNA and protein synthesis in Lewis lung carcinoma cells while slightly increasing their adherence to human microvascular endothelial cells [149]. In human colorectal adenocarcinoma cell HT29, OA both enhanced and inhibited store-operated Ca^2+^ entry (SOCE), which is associated with proliferation. At low concentrations (1 and 10 μM), OA increased SOCE, but at higher concentrations, OA potently inhibited it, suggesting that different concentrations of OA might trigger different mechanisms [115]. In TSCC, OA effectively inhibited cell proliferation in a dose- and time-dependent manner, which was associated with lower activation of specific downstream signaling pathways as indicated by the phosphorylation level of key proteins (p-Akt, p-mTOR, p-S6K, p-4E-BP1 and p-ERK1/2). In human esophageal cancer cells OE19 and OE33, OA downregulated cell proliferation, adhesion, and migration via the activation of tumor suppressor genes p27, p21, and p53. OA also increased AMPK phosphorylation but decreased p70S6K activation [150]. In TSCC cells CAL27 and UM1, OA treatment significantly induced cell cycle G0/G1 arrest and increased the proportion of apoptotic cells as shown by decreased expression of CyclinD1 and Bcl-2 and increased expression of p53 and cleaved caspase-3. OA also induced the formation of autolysosomes and decreased the expression of p62 as well as the LC3 I/LC3 II ratio [126]. A recent study investigated the effects of OA treatment in two hepatocellular carcinoma cell lines (Hep3B and Huh7.5) and in a healthy-liver-derived human cell line (THLE-2). OA treatment reduced cell migration and invasion. It also increased cell death by apoptosis and necrosis, while it had no effects on healthy cells [151]. However, the high concentration used (300 μM) and long exposure (48 h) raised concerns that the inhibition of cell migration and invasion could be due to OA-induced apoptosis [116]. In addition, OA potently inhibited telomerase activity, which plays an important part in the cellular immortalization of cancers [152]. Furthermore, OA and its metabolite, oleoylethanolamide, inhibited programmed death-ligand 1 (PD-1) expression and induced apoptosis via STAT phosphorylation in several cancer cell lines, namely A549, HuH-7, MCF-7, DLD-1, and LoVo cells [153]. In addition to OA, myristoleic acid (C14:1) extracted from *Serenoa repens* induced LNCaP prostate cancer cell death by apoptosis and necrosis [58]. It is also reported that both cis- and trans-vaccenic acid inhibited cancer cell growth [64,65,66], and erucic acid (C22:1) inhibited glioblastoma cell C6 proliferation, inhibiting DNA synthesis via PPAR activation [74,75].

Lastly, OA was found to potentially interact with cancer therapy agents. In BT-474 and SKBr-3 breast cancer cells, OA downregulated the expression of the Her-2/neu (erbB-2) oncogene, and concurrent exposure to OA and trastuzumab synergistically enhanced the growth inhibition effects of this chemotherapy drug [154,155]. Because of the pH responsiveness, newly developed OA-based nanostructures have the potential to efficiently target tumors, combining drug delivery with the therapeutic potential of OA. This could become a powerful strategy for the targeted treatment of metastatic melanoma [156]. A recent study found that MUFA radiosensitized cervical cancer cells through a novel p53-dependent mechanism. MUFAs activated PPARγ and p53 to promote lipid uptake, storage, and metabolism after radiotherapy [157]. Furthermore, OA interacts with some anti-cancer proteins such as α-lactalbumin and lactoferrins. For example, HAMLET (human alpha-lactalbumin made lethal to tumor cell), a molecular complex of human α-lactalbumin and OA, is known to have selective cytotoxic activity against certain types of tumors, and OA might play a key role in HAMLET-induced tumoricidal action [158,159,160]. In patients with advanced cancer, a combination of OA and Gc-protein-derived macrophage activating factor was shown to have a significant influence on immune system stimulation and the reduction in tumor mass while avoiding harmful side effects [161]. In addition, OA was found to increase the absorption of drugs by decreasing breast cancer resistance protein or P-glycoprotein mediated efflux [162,163].

**Figure 3 ijms-24-09921-f003:**
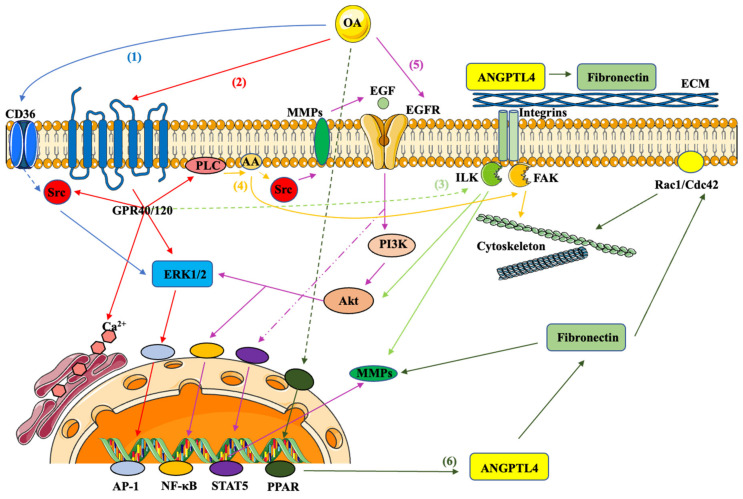
Overview of tumor-promoting signaling triggered by OA. (1) OA increases the expression of CD36, promoting cell growth and invasion via Src and ERK1/2. (2) OA promotes cell proliferation and migration via GPR40/120 activity, Ca^2+^ release from the ER and downstream activation of the ERK1/2/AP-1 signaling pathway. (3) OA promotes cell proliferation through a GPR40/ILK/Akt pathway and enhances cell invasion via ILK/MMP-9. (4) OA promotes cell migration through an arachidonic acid (AA)-dependent pathway associated with Src and FAK activation, while GPR and PLC are necessary for AA accumulation. (5) OA mediates EGFR transactivation (or Src/MMPs/EGF signaling), activating ERK1/2/NF-κB via the PI3K/Akt pathway. In addition, OA induces cell migration through a Stat5-dependent pathway with EGFR and MMPs involved. (6) OA promotes cell migration and invasion via an ANGPTL4/fibronectin pathway, activating MMPs, Rac1 and Cdc42. PPAR is involved in the activation of ANGPTL4 by OA. Dotted lines indicate indirect activation by mechanisms that remain to be clarified. The figure was created using Servier Medical Art image templates under a Creative Commons Attribution 3.0 Unported License.

## 3. Role of Endogenously Synthesized MUFA in Cancer

As the limiting step in MUFA synthesis, SCD activity (and its modulation using pharmacological inhibitors or by over/under-expression) provides insights into the impact of endogenously synthesized MUFA on cancer. The impact of SCD1 activity in cancer cells is summarized in Figure 4.

### 3.1. SCD Activity and Cancer—Evidence from Human Studies

Extensive clinical and epidemiological research has been performed to study the role of SCD in cancer as well as its association with cancer progression and death rates. A recent study evaluated the expression of *SCD1* in different cancer types utilizing The Cancer Genome Atlas database. Compared with normal tissues, *SCD1* expression was upregulated in most types of cancer including bladder urothelial carcinoma, cervical squamous cell carcinoma, colon adenocarcinoma, esophageal carcinoma, head and neck squamous cell carcinoma, kidney chromophobe, kidney renal clear cell carcinoma, kidney renal papillary cell carcinoma, and stomach adenocarcinoma. On the contrary, in thymoma, pheochromocytoma and paraganglioma, lung adenocarcinoma, glioblastoma multiforme, and breast invasive carcinoma, SCD1 was downregulated. Gastric cancer patients with higher SCD1 expression have relatively less-optimistic prognostic outcomes and relatively shorter overall survival [164]. A study performed SCD1 immunohistochemistry on 11 different tumors: breast, colon, lymphoid, prostate, gastric, ovary, brain, kidney, liver, skin, and lung. SCD1 expression was detectable in >75% of tumors tested, and more than 50% of tumors showed strong staining while corresponding normal tissues showed relatively low SCD1 expression [165]. Our laboratory also investigated the impact of SCD1 expression in metastatic breast cancers by generating Kaplan–Meier plots over a period extending up to 180 months by using available gene expression dataset records. These analyses reveal that high *SCD1* expression in the primary tumor is significantly associated with an increased proportion of metastasis-related deaths in patients suffering from breast cancers. Of note, this association appears even stronger in triple-negative cancer, as reflected by the elevated hazard ratio associated with this cancer subtype [146]. An exploratory study measuring *SCD1* expression levels in primary tumors also found a higher expression level in HER2+ and HR+ breast cancers. In this study, *SCD1* expression was associated with shorter relapse-free survival and shorter overall survival by multivariable analysis [166]. Another group performed immunohistochemical staining of a tissue microarray containing a total of 192 cores from different breast cancer subtypes. SCD1 expression was higher in cancer tissues compared with normal adjacent breast tissue. The expression of SCD1 was also found to be correlated with tumor grade and was associated with low overall survival in patients [167]. The association between high SCD1 expression in cancer tissue samples and poor clinical prognosis was also observed in bladder cancer [168], prostate cancer [169], pancreatic cancer [170], ovarian cancer [171], lung cancer [172], colorectal cancer [173], clear cell renal cell carcinoma [174,175], and cervical cancer [176]. A Swedish men study also showed an association between single-nucleotide polymorphisms in the SCD-1 gene and cancer death [59]. However, studies on SCD5 expression in cancer drew different conclusions. An analysis of samples from public databases showed that SCD5 expression in different cancers could be either upregulated or downregulated. It was downregulated in breast cancer, and low expression of SCD5 was associated with more aggressive breast cancer phenotypes, such as high histological grade, late stage, and HER2 overexpression. Survival analysis revealed that there was no correlation between SCD5 expression and overall survival, while upregulated SCD5 expression was related to longer breast-cancer-free survival [177]. In addition, SCD5 was reported to be significantly higher in primary and low-invasive melanoma than in metastatic cell lines or in five independent cultures of normal melanocytes, at both the mRNA and protein levels [178]. However, compared with the widely studied SCD1, SCD5 has limited expression and is poorly characterized. More research is needed to fully understand its role in cancer.

In accordance with the abnormal expression of SCD, an unbalanced amount of SFA and MUFA has been observed in blood and tissue samples from cancer patients [179,180,181]. A study analyzed the FA composition of phospholipids in the membranes of red blood cells from cancer patients and found that these phospholipids have a lower percentage of SFA and a higher percentage of MUFA compared to controls [182]. Lipid imaging profiling of six different cancer types (breast, lung, colorectal, esophageal, gastric, and thyroid cancer) showed a significant increase in MUFA and monounsaturated phosphatidylcholine levels in the cancer microenvironment compared with the adjacent normal tissue [34]. A study on Swedish men showed an association between increased circulating palmitoleic acid in serum lipids and future cancer death [59]. Breast cancer studies reported higher levels of MUFA in blood/plasma lipids [183] and breast adipose tissues [184]. A prospective study also found that blood levels of MUFA were related to prostate cancer incidence, and this association was even stronger for high-grade (Gleason ≥ 7) tumors [36]. Compared to normal hepatocytes, SCD expression levels and the concentration of its MUFA products were increased in aggressive hepatocellular carcinoma and were associated with poor survival times and tumor recurrence [185]. Another study showed that the quantity of MUFAs in cholesterol esters was positively correlated with a higher patient death rate [35].

The studies discussed above presented elevated MUFA levels as detrimental to cancer patients. However, a few additional studies showed contradictory results. A nested case–control study examined the FA composition of erythrocyte membranes from prostate cancer patients and found no significant association between MUFA and cancer risk [186]. In addition, a higher MUFA:SFA ratio was positively associated with decreased colon cancer risk in the Singapore Chinese Health Study [187]. Moreover, genotype data from 9254 colorectal cancer cases and 18,386 controls of European ancestry allowed one research group to correlate predicted plasma MUFA levels and a reduction in the risk of colorectal cancer [188]. Statistically significant inverse associations were found between high plasma levels of MUFAs and the risk of pancreatic cancers from a nested case–control study in Iran [189]. A prospective analysis showed an inverse association for MUFA levels, especially OA, with T-cell non-Hodgkin lymphoma risk [190]. Furthermore, some breast cancer studies revealed an inverse association with cancer aggressiveness. Lipid composition quantification analysis from the tumors freshly excised from breast cancer patients found that MUFA concentrations, in lymphovascular invasion (LVI)-positive breast carcinoma, were significantly lower than those in LVI-negative tumors [191]. Fatty acid analysis from the mammary adipose tissue of postmenopausal women showed that patients with malignant lesions had significantly lower MUFA levels compared to those with benign disease and a history of breast cancer [192].

### 3.2. SCD Activity and Cancer—Evidence from Animal Studies

There are several potentially redundant SCD isoforms in mice (SCD1–4) [193], making it challenging to study SCD activity in these animal models. Furthermore, SCD depletion (expression and activity) causes phenotypes including dry skin, alopecia and sebocyte hypoplasia [32,194,195]. These phenotypes, though seemingly innocuous at first, become more severe with age and can lead to blindness. Consequently, genetically modified xenograft models (*SCD* gene knockdown or overexpression), as well as pharmaceutical inhibition of SCD activity, are commonly used to study the role of this enzyme family in tumor development and progression (see Table 3).

SCD1 plays a key role in mouse tumor growth. One study determined the role of SCD1 in lung tumor growth by the subcutaneous injection of SCD1-deficient and control A549 lung cancer cells into athymic nude mice. Compared with control, SCD1-deficient mice showed increased tumor latency and reduced tumor growth rate, with about 40% in contrast to 100% tumor formation in the control group [32]. Another study observed lighter and smaller tumors compared with controls through the implantation of SCD1-knockdown H1650 lung cancer cells in mice [172]. In addition, treatment with the SCD1 inhibitor A939572 in an A549 lung cancer xenograft mouse model attenuated tumor growth and showed enhanced anti-tumor activity in combination with amodiaquine (an anti-malarial drug) [196]. The tumor inhibition effects of A939572 were also documented in xenograft tumors of LOVO colorectal cancer cells [197], GA16, and (SCD1-overexpressing) MKN45 gastric cancer cells [164,165], Panc02 pancreatic cancer cells [198], and FT-t ovarian cancer cells [171]. In a C4-2 prostate cancer xenograft mouse model, treatment with BZ36, a specific pharmaceutical SCD1 inhibitor, also significantly reduced tumor volume and tumor growth rate. Interestingly, BZ36 treatment induced tumor regression and resulted in a significant and dose-dependent increase in life span in comparison with control mice [169]. Another group implanted DU145 prostate cancer cells expressing doxycycline-inducible shRNAs into the prostates of immunocompromised mice and found that SCD1 ablation resulted in a significant increase in life span and a substantial attenuation of tumor growth in the early doxycycline treatment regimen [167]. In line with this, ectopic expression of SCD1 facilitated tumor formation and growth in an SCD1-overexpressing LNCaP prostate cancer cell model [199] and MKN45 gastric cancer cell model [164]. In a bladder cancer study, tumor growth was suppressed by treatment with SCD1 inhibitor A37062 in UMUC-14 xenografts models and by doxycycline-inducible knockdown of SCD1 in an SW780 tumor model. These results were associated with a decrease in the ratio of MUFAs to saturated FAs in the tumors and liver tissues, as well as an induction of apoptosis as determined by caspase-3 cleavage [200]. Of note, the novel piperidine derivative SCD1 inhibitor T-3764518 showed dose-dependent growth inhibition of xenograft mouse models of HCT116 colorectal carcinoma, MSTO-211H lung mesothelioma carcinoma and 786-O renal cell adenocarcinoma [201,202]. This inhibitor showed great pharmacokinetic properties in mice according to oral absorption and tumor distribution. Lipidomic profile analysis revealed a lower desaturation index in T-3764518-treated mouse tumor tissues, suggesting efficient in vivo inhibition of SCD1 activity [202]. Interestingly, in a liver tumor xenograft mouse model, conditional knockdown of SCD2 in primary hepatic stellate cells (the major isoenzyme in these cells) significantly slowed tumor formation and development [203].

Furthermore, animal studies have shown that SCDs are involved in cancer metastasis. The incidence of lung metastasis decreased in mice having undergone tail vein injection of colorectal cancer cell HCT116 where *SCD1* was silenced as compared to controls. Histological analysis showed a decreased size and number of lung metastatic tumors following *SCD1* suppression [173]. Similar metastasis inhibition results were also reported in an SCD1-knockdown hepatocellular carcinoma model [204]. In a gastric cancer xenograft model, Twist1 (a key transcription factor driving metastasis) positive cells were found to overexpress *SCD1*, implicating it in the metastasis process [164]. Interestingly, in an SW1 melanoma mouse model, treatment with the SCD inhibitor A939572 inhibited tumor growth but promoted a substantial increase in lung metastases [205]. Another study on nude mice injected with SCD5-overexpressing A375M melanoma cells and 4T1 mammary carcinoma cells showed significantly less metastasis formation in the lung compared to the mice injected with control cells. Primary tumors derived from SCD5-overexpressing cells showed diminished fibrotic morphology and fewer extracellular bundles, suggesting reduced extracellular matrix (ECM) deposition, which is associated with fewer metastases [178]. In a 4T1 triple-negative breast cancer mouse model, SCD5 overexpression hampered metastatic spreading by blocking SPARC (secreted protein and rich in cysteine) secretion, which plays a role in decreasing ECM deposition and reverting the EMT [206].

**Table 3 ijms-24-09921-t003:** SCD activity in mouse cancer models.

Cancer Type	SCD Model	Mouse Model	Main Outcome	Reference
Bladder cancer	SCD1 knockdown, SCD1 inhibitor A37062	Xenograft of SW780, UMUC-14 cells	Inhibited tumor growth and progression	[200]
Breast cancer	SCD5 overexpression	Xenograft of 4T1 cells	Reduced tumor aggressiveness	[206]
Colorectal cancer	SCD1 inhibitor A939572	Xenograft of LOVO cells	Reduced tumor volume and tumor weight	[197]
Colorectal cancer	SCD1 knockdown	Tail vein injection xenograft of HCT116 cells	Decreased the size and number of lung metastatic tumors	[173]
Colorectal/Lung/Renal cancer	SCD1 inhibitor T-3764518	Xenograft of HCT116/MSTO-211H/786-O cells	Inhibited tumor growth	[201,202]
Gastric cancer	SCD1 overexpression, SCD1 inhibitor A939572	Xenograft of MKN45 cells	Overexpression of SCD1 enhanced proliferation and metastasis while inhibition reduced both tumor volume and tumor weight	[164]
Gastric cancer	SCD1 inhibitor A939572	Xenograft of GA16 cells	Inhibited tumor growth	[165]
Liver cancer	SCD1 knockdown	Xenograft of HepG2 cells	Inhibited tumor size and metastasis	[204]
Liver fibrosis	SCD2 conditional knockout, SCD inhibitor A939572	SCD2 conditional knockout	Reduced liver fibrosis, tumor formations, tumor size and tumor multiplicity	[203]
Lung cancer	SCD1 knockdown, SCD1 inhibitor A939572	Xenograft of A549 cells	Reduced tumor growth, fewer tumor formations and increase in tumor latency	[32]
Lung cancer	SCD1 knockdown	Xenograft of H1650 cells	Reduced tumor weight and volume	[172]
Lung cancer	SCD1 inhibitor A939572	Xenograft of H460 cells	Inhibited tumor growth	[196]
Melanoma	SCD1 inhibitor A939572	Xenograft of B16F1, SW1 cells	Inhibited primary tumors growth but increased lung metastases	[205]
Melanoma	SCD5 overexpression	Xenograft of A375M, 4T1 cells	Reduced metastases	[178]
Ovarian cancer	SCD1 inhibitor A939572	Xenograft of FT-t cells	Reduced tumor number and mass	[171]
Pancreatic cancer	SCD1 inhibitor A939572	Xenograft of Panc02 cells	Reduced tumor size	[198]
Prostate cancer	SCD1 inhibitor BZ36	Xenograft of LNCaP, C4-2 cells	Inhibited tumor volume and tumor growth rate	[169]
Prostate cancer	SCD1 knockdown	Xenograft of DU145 cells	Inhibited tumor growth	[167]
Prostate cancer	SCD1 overexpression	Xenograft of LN cells	Increased tumor formation and growth	[199]

### 3.3. SCD Activity and Cancer—Evidence from Cellular Models

Generally, high MUFA concentrations are a result of increased SCD activity. As such, elevated MUFA levels in cell membranes and the corresponding extracellular vesicles of PC3 human prostatic adenocarcinoma cells, as compared to the less aggressive LNCaP cells [207], imply enhanced SCD activity in the most aggressive cell line. However, SCD activity is not limited in its impact to cancer cell biology. For example, cancer stem cells contain a distinctive lipid profile, with higher free MUFA and lower free SFA levels than bulk cancer cells, which suggests that increased lipid desaturation is essential to stem-like characteristics in cancer cells [208,209].

#### 3.3.1. Role of SCD1 in Cell Proliferation and Cell Cycle

In H460 lung cancer cells, the use of the SCD1 inhibitor CVT-11127 significantly decreased cell proliferation, an effect that could be reversed by the addition of exogenous MUFAs (OA, palmitoleic or cis-vaccenic acid) [60,210]. It was also reported that the population of H460 cells in the S phase was decreased by almost 75% with a concomitant increase in the G_1_ phase following treatment with CVT-11127, with no changes in the G_2_/M phase. However, a 50% decrease in the G_2_/M phase was observed in cells exposed to the SCD inhibitor in serum-deficient media. This indicated that MUFA-containing lipids in serum possibly sustained the passage of SCD1-deficient cells through mitosis. Exogenous OA reversed the cell cycle changes induced by SCD1 inhibition, confirming that SCD1 impacts cell cycle progression through its MUFA product [60]. In A549 lung cancer cells where *SCD1* was suppressed, the MUFA/SFA ratio in total lipids lowered, and cell proliferation and growth were considerably decreased [32]. Interestingly, two different SCD1 inhibitors were found to suppress cell growth in A549 lung cancer cells, but only following EGFR activation. Further analysis found that SCD1 phosphorylation on Y55 by EGFR kinase activity was critical for it to enhance lung cancer growth [211]. In HeLa cervical cancer cells, *SCD1* knockdown was found to decrease cell proliferation and reduce colony formation ability [176]. In line with this, *SCD1* overexpression in HEK293 cells led to the significant promotion of colony formation while cell growth and colony formation were inhibited in H1650 cells where *SCD1* was suppressed [172]. In SW780 and UMUC-14 bladder cancer cells, *SCD1* knockdown by siRNA inhibited cell proliferation in an FA-desaturation-dependent manner while this effect was reversed by the exogenous addition of OA [200]. In three different bladder cancer cell lines (UMUC-14, TCC-97-7 and SW780), DNA synthesis suppression was observed following *SCD1* knockdown. Further analyses in SW780 cells revealed a reduced percentage of cells in the G_2_ and S phases and an increased percentage in G_1_ 48 h after *SCD1* knockdown [200]. In LNCaP and C4-2 prostate cancer cells, the inhibition of SCD1 activity by BZ36 induced a dose-dependent decrease in cell proliferation, reaching 100% inhibition at the maximal dose used. Flow cytometry analysis showed an accumulation of LNCaP cells in the G_0_/G_1_ phase of the cell cycle and a decrease in the S phase upon BZ36 treatment. However, no effect on proliferation was observed in the non-cancerous prostate cell line PNT2, even at a maximal dose. Similarly, *SCD1* knockdown resulted in a decrease in the proliferation while SCD1 overexpression increased cell proliferation in both LNCaP and C4-2 cells [169].

#### 3.3.2. Role of SCD1 in Cell Migration and Invasion

It was reported that *SCD1* expression increased cell membrane fluidity as well as fibroblast-induced EMT and migration in poorly (MCF-7) and highly (MDA-MB-231) invasive breast cancer cells. The inhibition of SCD1 by siRNA and inhibitor A939572 both resulted in a significant inhibition of MCF-7 and MDA-MB-231 cell migration promoted by fibroblast-released soluble factors [212]. Further study showed that the effects of SCD1 inhibition could be rescued by the addition of OA [213]. Our laboratory also reported that SCD1 activity was implicated in the transformation of MDA-MB-231 cells from an epithelial to a mesenchymal phenotype. Silencing *SCD1* was associated with increased GSK3 activity, a reduction in β-catenin nuclear localization and transactivation activity. It also modified cell shape and their invasive potential by the reduction of cell spreading and cell–cell junctions [214]. In MDA-MB-231 cells, the inhibition of SCD1 induced a rounded, less elongated phenotype, while OA treatment resulted in elongated, spindle-shaped cells, which was associated with increased speed and diminished directional changes during migration [146]. In HeLa cervical carcinoma cells, *SCD1* knockdown significantly inhibited cell migration and invasion abilities in wound healing and Transwell assays, and the expression level of EMT-related proteins was decreased [176]. In 786-0 clear cell renal cell carcinoma cells, the depletion of *SCD1* diminished cell migration and colony formation ability [215]. In HCT116 and SW116 colorectal cancer cells, stable *SCD1* knockdown impaired migration and invasion ability while ectopically expressed SCD1 in Caco2 cells significantly increased migration and invasion rates. These effects were associated with increased MUFA levels and the suppression of PTEN expression and Akt activity [173]. Similarly, the overexpression of SCD1 in HEK293 cells promoted cell invasion and migration while the knockdown of *SCD1* in lung cancer cell H1650 had opposite effects [172]. However, the expression of SCD5 was lower in more aggressive metastatic breast cancer and melanoma cells than primary breast cells and low-invasive melanoma [177,216], and supplementation with OA reduced A375M melanoma cell malignancy by reducing the dissemination capability, impairing tumor spread [178]. In MDA-MB-231 cells, cell migration was not significantly affected by siRNA-mediated SCD5 depletion [213].

#### 3.3.3. Role of SCD1 in Cell Death

Studies have reported that the inhibition of SCD1-induced ER stress and cell death through several mechanisms, including apoptosis and ferroptosis. In SW780 and UMUC-14 bladder cancer cells, *SCD1* knockdown significantly increased the levels of the apoptotic cell-surface marker Annexin-V. It was also associated with the cleavage of caspases-3 and -7 and activation of the caspase-3 substrate PARP suggesting a stimulation of apoptosis [200]. In U2OS and SW480 colorectal cancer cells, *SCD1* depletion induced high-level induction of caspase-3 activity and PARP cleavage as well as unfolded protein response hallmarks such as Xbp1 mRNA splicing, phosphorylation of eIF2α and increased expression of the apoptosis-related protein C/EBP homologous protein [217]. In Caki1 and A498 clear cell renal cell carcinoma cells, both genetic knockdown and pharmacologic inhibition of SCD1 decreased tumor cell proliferation and induced apoptosis. The induction of ER stress response signaling was also observed upon the inhibition of SCD1 activity (A939572) while OA treatment reversed these effects [174]. A recent study performed in ovarian cancer cells showed that SCD1 depletion or inhibition lowered MUFA levels and triggered the ER stress response with the activation of the IRE1α/XBP1 and PERK/eIF2α/ATF4 pathways. The induction of long-term mild ER stress or short-time severe ER stress led to cell death by apoptosis, and supplementation with OA rescued these effects [218]. Furthermore, in an ovarian cancer study, the inhibition of SCD1 by inhibitors MF-438 or CAY10566 and *SCD1* knockdown reduced cell viability and increased cell death. This was restored by providing cells with either SCD1’s product OA or the ferroptosis inhibitor Fer-1. In addition, cell death triggered by the ferroptosis inducer RSL-3 could be rescued by MUFAs (palmitoleic acid or OA) but not by SFAs (palmitic or stearic acid) [171]. A recent study found that the transcription factor BTB and CNC homology 1 (BACH1) induced ferroptosis in esophageal squamous cell carcinoma cells by negatively regulating *SCD1* through binding to its intron-2 region. *SCD1* knockdown significantly increased lipid ROS accumulation in KYSE150 and KYSE170 cells, while OA addition significantly attenuated oxidative stress and ferroptosis [142]. In MKN45 and HGC27 gastric cancer cells, SCD1 overexpression enhanced anti-ferroptosis markers SLC7A11 and GPX4. SCD1 overexpression also prevented Erastin-induced ferroptotic cell death and characteristic lipid oxidation [164]. Interestingly, a study showed that depletion of SCD5 in MCF-7 breast cancer cells induced necrosis. The double *SCD1* and *SCD5* knockdown did not worsen cell viability compared to single *SCD5* silencing. This necrotic effect was rescued by a 48 h treatment of cells with OA, suggesting that SCD5 maintains cell survival via the production of OA [213].

**Figure 4 ijms-24-09921-f004:**
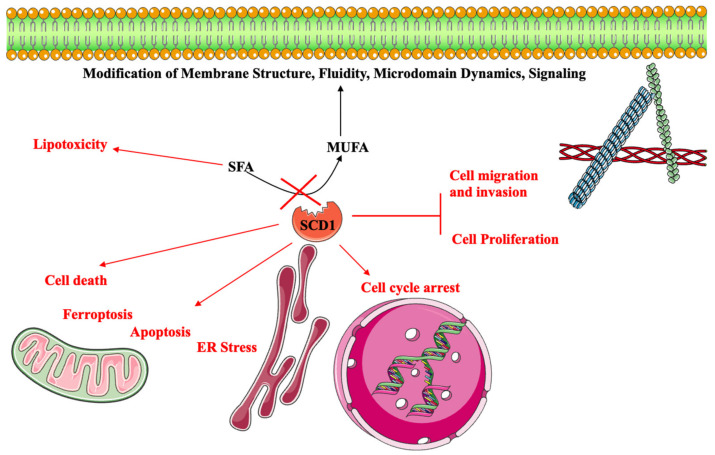
Impact of stearoyl-CoA desaturase 1 activity in cancer cells. Stearoyl-CoA desaturase 1 (SCD1) modifies cell membrane structure, membrane fluidity, microdomain dynamics and signaling through synthesis of monounsaturated fatty acids (MUFAs). Inhibition of SCD1 increases the SFA/MUFA ratio, causing lipotoxicity, ER stress, and cell death. SCD1 inhibition diminishes tumor cell proliferation, migration, and invasion. Black lines indicate the consequences of SCD1 activity in cancer cells while red lines indicate the anticancer effect of SCD1 inhibition.

## 4. Conclusions and Future Perspectives

The exploration of MUFA and the key SCD enzymes in the context of cancer research is a relatively recent area and is still evolving. The numerous studies discussed here highlight the intricate relationship between MUFA metabolism in the regulation of cancer development and progression. Regarding dietary MUFA, the results from epidemiologic and animal studies are inconclusive. Largely influenced by the well-known Mediterranean diet, some epidemiological studies have reported a protective role for dietary MUFA consumption in breast, colorectal, digestive, pancreatic, and prostate cancers. However, there are some contradictory findings suggesting no association in breast, colorectal and pancreatic cancers, and even increased risks in colorectal, liver, oral, and pancreatic cancers. In a similar contradictory fashion, animal studies based on MUFA-enriched diets have shown an inhibitory effect on tumor growth in breast, colorectal, and lung cancers while tumor-promoting effects were observed in cervical and pancreatic cancers. Many factors, including the dietary source, MUFA type, and conformations (cis-/trans-FA), could contribute to the controversial relationship between MUFA intake and cancer risk. MUFAs derived from plant sources, such as extra-virgin olive oil, have been negatively correlated with cancer risk. While MUFAs from animal sources, such as meat, seem to have more adverse effects. Interestingly, MUFAs that are derived from trans-fatty acids such as elaidic acid, are also positively correlated with cancer risk. In addition, the substantial studies on SCD1, the most well-characterized enzyme implicated in MUFA synthesis, have shown a strong correlation between its expression and activity and the development of a variety of cancer types, including breast, bladder, cervical, colorectal, esophageal, gastric, lung, ovarian, pancreatic, and prostate cancers. Deprivation of SCD1 has been shown to have antiproliferative and pro-apoptotic effects in both animal and cellular studies. In accordance with aberrant SCD1 activity, imbalanced MUFA levels have been observed in various cancer patients. Of note, a series of key players in cell signaling, including GPR, Ca^2+^, PKC, ERK, EGFR, MMP, PI3K/Akt and PLD2/mTOR, have also been implicated in OA-induced cancer cell proliferation, migration, invasion, and survival.

As a result, SCD1 appears to be a promising potential target for cancer therapy. Several SCD1 inhibitors (including A939572, CAY10566, MF-438, CVT-11127, and T-3764518) have already been tested as anticancer agents in different cancer models, both in vivo and in vitro. These SCD1 inhibitors slow cancer development and progression by inducing cell death as well as inhibiting angiogenesis [82,83,84,85,88,90,91,95,98,101,107,111,219]. These inhibitors can also improve chemotherapy and radiation therapy responses by reducing inflammation, oxidative stress and insulin resistance as well as enhancing the efficacy of other cancer therapeutic agents [80,103]. Although the results of the preclinical SCD1 inhibitor studies are promising, inhibiting SCD1 could disrupt lipid metabolism, potentially affecting normal cellular functions and leading to side effects. In fact, severe adverse effects have been observed in animal studies, such as eye and skin dryness, hair loss and cold-induced hypothermia [92,93,94,95], which is the primary challenge preventing these inhibitors from being applied to cancer therapy. Therefore, new strategies are needed before SCD1 inhibitors can be fully translated into clinical trials. From this perspective, an intriguing alternative would be to use dietary MUFAs to potentially overcome the side effects of SCD1 inhibition, as cancer cells are more dependent on SCD1 activity than normal cells. By incorporating dietary MUFAs, it might be possible to compensate for the reduced endogenous production of MUFAs to some extent, maintaining lipid homeostasis as well as supporting membrane stability and function. In addition, MUFAs, particularly those found in olive oil and avocados, have been associated with anti-inflammatory and antioxidant properties [220,221], which potentially alleviate the inflammatory and oxidative damage caused by SCD1 inhibition. Interestingly, newly developed OA-based nanostructures showed the potential to efficiently target tumors [142,197,222,223] and could be used to deliver SCD1 inhibitors. Thus, the combination of dietary modifications of MUFA intake and SCD1 inhibitors targeting MUFA synthesis hold promise as powerful approaches to cancer therapy. Ongoing endeavors to identify and optimize these inhibitors are necessary to determine the optimal dose and combination strategies to overcome side effects and improve efficiency. Conducting well-designed clinical trials across different cancer types can also provide valuable data on the potential benefits and limitations of targeting SCD and modulating MUFA metabolism in cancer patients.

Notably, there are still unresolved questions regarding the role of MUFA metabolism in cancer biology. Firstly, while the impact of dietary MUFAs on cancer in vivo is debated, exploring dietary interventions for cancer prevention and therapy is potentially valuable. However, controlling for confounding factors in epidemiological studies is challenging, making it difficult to isolate the specific effects of MUFA intake on cancer risk. Animal studies using purified MUFAs offer better control and focused investigations. Secondly, understanding the role of SCD and MUFAs in individual tumors and patient responses to treatment can pave the way for personalized therapeutic approaches. Identifying specific molecular biomarkers and genetic characteristics associated with SCD dysregulation or MUFA metabolism may also help identify patient subgroups that are more likely to respond to SCD inhibition or benefit from dietary modifications involving MUFAs. In addition, the existing understanding of the influence of MUFA on cancer primarily stems from studies focused on SCD1 and OA. However, recent studies have revealed alternative fatty acid desaturation pathways independent of SCD1 activity in cancer cells, involving unconventional MUFAs and desaturases [50,177,222,224,225]. Recent investigations into SCD5 in metastatic melanoma cells have shown distinct expression patterns and roles in cancer progression, which is contrary to SCD1 [32,194,195]. These discoveries could explain the contradictory results of cancer cells reacting to SCD1 deprivation to some extent. Future studies on these less common desaturases, such as SCD5 and FADS2, as well as on other MUFAs, such as palmitoleate and its isomers, could lead to new strategies targeting MUFA metabolism in cancer therapy. Lastly, how exogenously supplemented MUFA and endogenously synthesized MUFA from desaturases act differently on cancer development and progression remains to be clarified. Consequently, further studies are warranted to expand our knowledge in these areas and gain a more comprehensive understanding of the effects of different desaturases and MUFAs on cancer. 

## Figures and Tables

**Figure 2 ijms-24-09921-f002:**
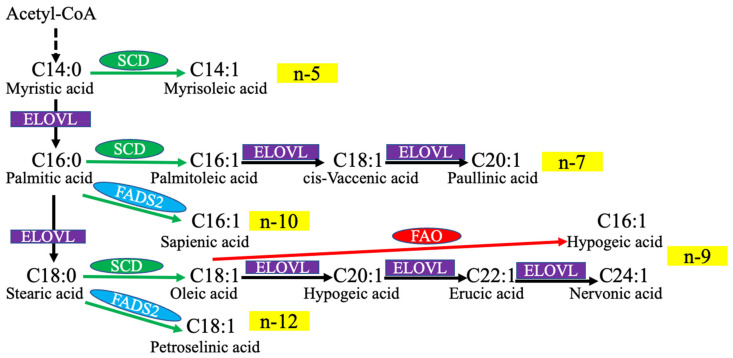
Biosynthetic pathways of main cis-monounsaturated fatty acids. SCD, stearoyl-CoA desaturase (1 and 5); FADS2, fatty acid desaturase 2; FAO, fatty acid oxidation; ELOVL, fatty acid elongase. Black arrows indicate fatty acid elongation. Green arrows indicate fatty acid desaturation. The red arrow indicates fatty acid oxidation. The figure is modified from [55,56].

**Table 1 ijms-24-09921-t001:** Summary of common MUFAs and effects on cancer.

MUFA		Source	Main Outcome	Reference
Myristoleic acid	14:1 (n-5), cis	Small amounts in nutmeg and nutmeg butter	Anti-cancer effects in prostate cancer cells	[58]
Palmitoleic acid	16:1 (n-7), cis	Nuts, meats, animal fats	Related to cancer death and rescued SCD1 blockade anti-cancer effects	[32,59,60,61]
Hypogeic acid	16:1 (n-9), cis	Human milk	Limited studies	[61,62]
Sapienic acid	16:1 (n-10), cis	Human sebum	Increased in lung and liver carcinomas and contributed to SCD inhibition resistance	[57,61,63]
cis-Vaccenic acid	18:1 (n-7), cis	Sea buckthorn oil	Inhibited colon cancer cell growth	[64]
Vaccenic acid	18:1 (n-7), trans	Human milk, dairy products	Inhibited cancer cell growth and proliferation and induced apoptosis	[65,66]
Paullinic acid	20:1 (n-7), cis	The seed oil of the plant *Pangium edule*	Limited studies	[67]
Oleic acid	18:1 (n-9), cis	Vegetable oils, such as olive oil, rapeseed oil and sesame oil	Both cancer-promoting and anti-cancer effects	See Section 2.3
Elaidic acid	18:1 (n-9), trans	Small amounts in caprine, bovine milk and some meats	Promoted survival, growth, and invasion of the colorectal cancer cell lines	[65,66,68]
Petroselinic acid	18:1 (n-12), cis	Several animal and vegetable fats and oils	Limited studies	[69]
Gondoic acid	20:1 (n-9), cis	Plant oils and nuts, such as jojoba oil	Limited studies	[70,71]
Gadoleic acid	20:1 (n-11), cis	Some fish oils, such as cod liver oil	Limited studies	[71,72]
Erucic acid	22:1 (n-9), cis	Brassica seeds, Indian mustard, rapeseed	Anti-cancer activity in brain cancer and glioblastoma	[73,74,75]
Brassidic acid	22:1 (n-9), trans	Seeds of certain brassica crops, such as mustard, rapeseed and kale	Limited studies	[72]
Nervonic acid	24:1 (n-9), cis	Animal brain, plant seed oil	Limited studies	[71,76,77]

## Data Availability

Not applicable.

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
