# Peer review of "Unveiling the MUFA–Cancer Connection: Insights from Endogenous and Exogenous Perspectives"

_ijms, 2023, doi:10.3390/ijms24129921_

Round 1

Reviewer 1 Report

The authors present an interesting bibliography compilation about the involvement and effects of MUFA and SCD in cancer. The work as merit, the research was thorough, the theme is contemporary and the work is well framed in the scope of a Special Issue about "Nutrition, Obesity and Metabolic Syndrome". Bibliographical collection was extensive and thorough, very up to date. The truth is that, with the data presented, the role of MUFA and SCD in cancer remains extremely dubious, but that is what the state-of-the-art in the issue brings us to and it does not necessarily remove merit from the work. The manuscript is extensive and not necessarily an easy read, relying many times on a necessary reader’s knowledge about mechanistic aspects and pathways of cancer at the cellular level. Therefore, although this work will probably especially appeal for people working in cancer biology, it should represent, because of its depth in terms of compiled bibliography used, a valuable resource to researchers in the field. There are some issues with the paper, but I believe they are manageable and solvable; therefore, I find the paper interesting enough and suitable for publication after the revisions proposed.

Main comments:

My main point of concern is the Conclusions section. Most of it is a repetition of studies and concepts detailed and discussed before in the manuscript, with an untimely digression about olive oil and its components that should not be important here. From people having done such a commendable effort dissecting and compiling information from such a large pool of manuscripts, working in the field, I would expect a more substantiated and opinionated composition, and essentially something in more general terms. With so many apparently incoherent reports, is this a line of work worth pursuing? So, are MUFA and SCD activity good or bad in terms of their involvement in cancer? (of course the answer is nuanced and it may go both ways, but a general appraisal with general arguments going each way could be presented) What seems more interesting to pursue right now in terms of cancer prevention in the authors’ point of view: dietary interventions or SCD modulation? What way are the most recent studies going, tendentially? Where do you see research in the subject going? Are more epidemiologic studies warranted? Or is it more important to understand the effect of MUFA more mechanistically, at this point? Basically, instead of referencing the work of others, the conclusion should integrate some expert opinion on the subject, and some future perspectives as well.

Minor comments:

Line 17: “…to provide new insights on MUFAs in cancer biology.” Consider rephrasing like: “…to provide new insights on MUFAs effects in cancer biology.” Or “…to provide new insights into MUFAs modulation/interference in cancer biology.”

Line 34: do you mean “hormone imbalance”?

Line 34: What is the effect on growth factor signalling? Interference? Dysregulation?

Line 45: “Lipids are a heterogeneous group of biomolecules that not only constitute the structural basis of biological membranes but also function as signaling molecules and energy sources.” Please add a reference.

Line 47: “The lipid class of molecules comprises fatty acids (FA), acylglycerols, phospholipids (PL), sterols, and sphingolipids 11, 14, 15.” The phrase should star with “In humans/mammals… Other organisms may present other lipid classes as well.

Line 84: “SCD activity…” I was always told, as a general rule, not to use abbreviations in the beginning of sentences. Please check the rest of the manuscript.

Line 99: “…which in turn releases very-low-density lipoproteins (VLDL) into the circulation where they are hydrolyzed into intermediate-density lipoproteins (IDL) and low-density lipoprotein (LDL).” Since this is in fact depicted in Figure 1A it should be better explained. How are VLDL transformed into IDL and LDL?

Line 200: “Dietary MUFA was also reported to be associated with an increased risk 200 of pancreatic cancer in a case-control study from the San Francisco Bay and a large pro-201 spective cohort from NIH-AARP Diet and Health Study that 74, 75.” That “that” at the end. Do not know if you would want to remove it or something additional was lost in the sentence.

Line 452: “Newly developed pH sensitive oleic acid-based nanostructures were reported combine drug delivery with the antimetastatic potential of OA.” I do not understand the sentence. Please rephrase.

Line 639: “High MUFA concentrations follow from increased SCD activity.” Please rephrase. ”Generally, high MUFA concentrations are a result of increased SCD activity.” Is this what you mean?

English in the manuscript is generally adequate. There are some problems at times, mostly with sentences that are not that clear, but I included those in my suggestions.

Reviewer 2 Report

1.     The manuscript “Monounsaturated Fatty Acids and Cancer: From Epidemiology to Cell Biology” covered MUFA and desaturase (SCD) on cancer progression. The authors primarily aimed to explore the assessment of MUFA on cancer as a therapy and risk factor. The article's title may be unique, but not the content regarding its concept, narrative and research direction in emerging areas of cancer and lipid metabolomics.

 2.     After carefully reading the manuscript and thoroughly reading the citations, it is concluded that the collection of cited cancer-related works is obtained from several in vitro cell line data. Eventually, the authors missed several recent developments in fatty acid's role in cancer metabolism. Some of these emerging concepts of Fatty acids and Cancer Metabolism are available here:

·       Reprogramming of fatty acid metabolism in cancer

https://doi.org/10.1038/s41416-019-0650-z

·       Fatty acids and evolving roles of their proteins in cancers

https://doi.org/10.1016/j.plipres.2021.101116

·       Evidence for an alternative fatty acid desaturation pathway increasing cancer plasticity

DOI: 10.1038/s41586-019-0904-1

3.     Despite the fact cancer cells use glucose as primary energy (Warburg’s hypothesis) over fats, cancer cells reprogram fatty acid metabolism in support of their rapid membrane formation and growth, which is evidenced by higher expression of fatty acid binding or carrier proteins in several types of cancers. The authors failed to include the roles of fatty acids in cancer metastasis in general. For example, palmitate is desaturated to the unusual fatty acid sapienate to support membrane biosynthesis during proliferation in cancer (see recent review work listed above)

4.     Unlike in vivo systems, where fatty acids undergo a series of steps from uptake, transport, metabolism, and absorption at specific concentrations, fatty acids are used as growth factors that cancer cells use to proliferate in vitro. Thus, cancer risk should not be viewed in terms of selective fatty acids but focus on a control point, which in this review is SCD. Thus, the author must revise the title accordingly and include a larger perspective (see the reviews) of fatty acid metabolic control (SCD) on cancer risk rather than a type of fatty acid like MUFA.

5.     The review missed distinguishing between nutrient and control points in cancer. Data are emerging that dietary fat composition has a greater impact on cancer pathogenesis than the total fat content. Lipid desaturation is an essential control point that promotes stem-like characteristics in type cancer cells. Thus, the study title and content have a little connection up to Line 91.

6.     Line 42 "New cancer treatment strategies are expected to emerge from these studies" The statement is supported by citations no. 12 and 13, which are not updated beyond 2013. The authors missed including several recent studies on the fatty acid metabolism of cancer cells.

7.     The endogenous production of MUFA is transient since these metabolites are rapidly converted to longer versions, and it is different in cancer cells. MUFAs serve as unsaturated fatty acid substrate pools that convert to 20- and 22-carbon n-9 PUFAs, further desaturation and elongating in cancer cells. The elongation capacities for unsaturated fatty acids use alternative fatty acid desaturation pathways, increasing cancer plasticity.

8.     It would primarily point towards MUFA from animal fat since MUFA from plant sources is rarely reported with toxic effects. Animal fats and a variety of seeds also have high MUFA content. In the case of MUFA, animal MUFAs have different effects on cancer cells than plant MUFA since the latter is rich in antioxidants. Unlike oleic acid, elaidic acid (trans-MUFA) has opposite effects on the cancer microenvironment cell membrane. Plant MUFAs are rarely associated with toxic effects, even heating high while used as oil.

9.     Fig.1 The purpose of the figure needs to be understood. The readers already know dietary lipid uptake and lipid metabolism.

10.  Fig.2 The narrative "Overview of tumour-promoting signalling triggered by OA" does not match evidence in the text where OA is protective alone or adjuvant with several anticancer regimens (lines 456-465)

11.  The conclusion section reads as an elaboration of the narrative review but does not contain a takeaway message for the readers.

12.  The work primarily focused on fatty acids metabolic control points like desaturase (SCD) and MUFA and their roles in cancer. The manuscript “title” must be revised to include all these aspects.

13.  Overall, the conceptualization of the review is relatively weak, lacks novelty, has inadequate citations and lacks a strong hypothesis. The structural aspect that needs to mention the tools used to construct the review must be included.

none

Reviewer 3 Report

This is a very interesting review, which provides outstanding information on monounsaturated fatty acids and their application in vitro cancer models, with encouraging results.

Round 2

Reviewer 2 Report

The revised version of the article has incremental improvement, while the conceptualization of the elaboration on fatty acids, including MUFA or Omega-9 fatty acids concerning cancer yet to achieve its stand as claimed by the authors.

Related studies have reported omega-9 fatty acids, a primary mono-unsaturated fatty acid (MUFA), which are not included.

Unlike other macronutrients, lipids and fats are metabolized slowly due to their stepwise uptake, transport, and absorption. Now the title is revised to include the term "metabolism", a process involving precursor and end product of fatty acids.

It is apparent from the response that authors would consider MUFA as an entity of start point rather than fatty acid per se; in that case, "metabolism" may not be appropriate.
